Readiness of the primary health care units and associated factors for the management of hypertension and type II diabetes mellitus in Sidama, Ethiopia

Mulugeta Tigist Kebede 1 2 tgmaedot@gmail.com
Kassa Dejene Hailu 2
1 School of Nutrition, Food Science and Technology, Department of Human Nutrition Hawassa University , Hawassa , Ethiopia
2 School of Public Health, Department of Public Health Hawassa University , Hawassa , Ethiopia
Valentim-Silva João Rafael
Electronic publication date: 2022 Aug 25
Publication date: 2022
Volume: 10
Electronic Location ID: e13797
Received 2022 Apr 29; Accepted 2022 Jul 6
Copyright: © 2022 Mulugeta and Kassa
Copyright year: 2022
Copyright holder: Mulugeta and Kassa
License: This is an open access article distributed under the terms of the Creative Commons Attribution License, which permits unrestricted use, distribution, reproduction and adaptation in any medium and for any purpose provided that it is properly attributed. For attribution, the original author(s), title, publication source (PeerJ) and either DOI or URL of the article must be cited.
License URL: https://creativecommons.org/licenses/by/4.0/

Keywords: Readiness

Funding: The authors received no funding for this work.

==============================
Background

In low-income nations such as Ethiopia, noncommunicable diseases (NCDs) are becoming more common. The Ethiopian Ministry of Health has prioritized NCD prevention, early diagnosis, and management. However, research on the readiness of public health facilities to address NCDs, particularly hypertension and type II diabetes mellitus, is limited.

Methods

The study used a multistage cluster sampling method and a health facility-based cross-sectional study design. A total of 83 health facilities were evaluated based on WHO’s Service Availability and Readiness Assessment (SARA) tool to investigate the availability of services and the readiness of the primary health care unit (PHCU) to manage type II diabetes and Hypertension. Trained data collectors interviewed with PHCU head or NCD focal persons. The study tried to investigate (1) the availability of basic amenities and the four domains: staff and guidelines, basic equipment, diagnostic materials, and essential medicines used to manage DM and HPN, (2) the readiness of the PHCU to manage DM and HPN. The data were processed by using SPSS version 24. Descriptive statistics, including frequency and percentage, inferential statistics like the chi-square test, and logistic regression models were used to analyze the data.

Results

Of the 82 health facilities, only 29% and 28% of the PHCU identified as ready to manage HPN and DM. Facility type, facility location, presence of guidelines, trained staff, groups of antihypertensive and antidiabetic medicines had a significant impact (P < 0.05) on the readiness of the PHCU to manage HPN and DM at a 0.05 level of significance. Facilities located in urban were 8.2 times more likely to be ready to manage HPN cases than facilities located in rural (AOR = 8.2, 95% CI [2.4–28.5]) and P < 0.05.

Conclusion and recommendation

The results identified comparatively poor and deprived readiness to offer HPN and DM services at lower-level health facilities(health centers). Equipping the lower-level health facilities with screening and diagnostic materials, essential medicines, and provision of basic training for the health care providers and NCD guidelines should be available, especially in the lower health care facilities.

Introduction

NCDs are chronic diseases resulting from genetic, physiological, environmental, and behavioral factors (Bishu et al., 2019). The World Health Organization (WHO) projections showed that in 2016 a total of 56.9 million deaths occurred worldwide. The deaths caused specific mortality due to NCDs accounted for 40.5 million (71%). When we see age-standardized death rates, 29.8 million deaths in people less than 70 years of age, an estimated 17.0 million (57%) of the death were due to NCDs (Bennett et al., 2018). This result shows that globally, the leading cause of mortality and premature death is occurring due to NCDs with preventable risk factors such as tobacco use, physical inactivity, unhealthy diet, and the harmful use of alcohol. The diseases require lifetime management and pose life-threatening problems when not controlled (Wild et al., 2004; Guariguata et al., 2014). Both diseases have societal and economic consequences due to their complications for a country, society, and the individual (Bennett et al., 2018; Karashima, 2014).

Globally, NCDs are causing a significant health impact on humans, especially in adulthood. Hypertension and diabetes mellitus (DM) are common chronic diseases that quickly increase in prevalence (Guariguata et al., 2014). These diseases contribute to cardiovascular diseases (CVD), accounts for 17 million (31%) deaths globally. Seventy-five percent of all CVD associated deaths occur in developing countries (Omar et al., 2020; Russell et al., 2019).

Worldwide, the burden of diabetes is associated with metabolic risks (disproportionate between weight and height) and behavioral factors (i.e., diets lower in fruits and vegetables, smoking, and sedentary lifestyle). In 2017, weight and height were disproportionately responsible for 30.8% of deaths and 45.8% of disability adjusted life in years (DALYs); dietary risk was responsible for 24.7% of deaths and 34.9% of DALYs (Naghavi et al., 2015; Watkins et al., 2016).

In 2010, an estimated 1.33 billion (1.32–1.34 billion) individuals had hypertension, with 346 million (336–356 million) in high-income nations and 985 million (977–994 million) in low- and middle-income countries.

Between 2000 and 2010, the age-standardized prevalence of hypertension increased by 2.5%. The number of people with hypertension increased by 354 million from the four types of NCDs (diabetes, hypertension, chronic obstructive pulmonary disease, and cancer). And whose prevalence rates noted to be increasing were hypertension and diabetes mellitus (African Regional Health Report, 2014).

In Africa, the prevalence of NCDs is rising rapidly. It is projected to cause almost three-quarters as many deaths as communicable, maternal, perinatal, and nutritional diseases by 2020, exceeding the most general causes of death by 2030 (Bennett et al., 2018).

According to community-based studies, diabetes and hypertension prevalence among adults in Ethiopia ranged from 2.0 percent to 6.5 percent and 9.3% to 30.3%, respectively (Guariguata et al., 2014; Dong et al., 2019). Approximately 42.3% of DM patients have hypertension as a comorbidity (Ejeta et al., 2021). However, the overall prevalence of poor glycaemic control and uncontrolled hypertension in patients attending public health facilities was 45.2% and 48.6%, respectively (Nigussie et al., 2021; Aberhe et al., 2020). The growth in prevalence of hypertension-DM comorbidity demands appraisal of facilities’ readiness to manage both hypertension and DM diseases.

Ethiopia is one of the highly populated African countries next to Nigeria, with an estimated population of 114,963,588 by 2020 (Haque et al., 2020). This contributed a lot to the acquirement of communicable diseases. However, in recent years, the rise of infectious diseases is changing its place with NCDs (Shiferaw et al., 2018). In the country, NCDs are estimated at around 42%. Among these, 27% of premature deaths occur before 70 years of age. Disability adjusted life years (DALYs) due to NCDs in the country has increased from 20% in 1990 to 69% in 2015 (Girum et al., 2020), which means the risk of dying from NCDs is increased two folds than that of the communicable, maternal, neonatal, and nutritional problems together (Tesema et al., 2020).

Regarding the preparedness of PHCU to manage DM and HPN a study conducted in Nepal, Peru, and Mozambique, all three countries, lack of functioning equipment and supplies, diagnostic tests are not available, and the unavailability of medications and laboratory tests will raise the patient’s out-of-pocket expenses when traveling to other health facilities. Essential types of equipment like blood pressure monitoring apparatus and glucometers with their respective consumables were not available or out-of-date. The same study revealed that there are no clear referral procedures and transportation to the PHCU, a lack of tailored guidelines, and insufficient information, education, and communication materials (Cárdenas et al., 2021; Biswas et al., 2018).

According to a service availability and readiness assessment (SARA) survey conducted in 2016, about 54% of all health facilities had the general service readiness index in Ethiopia, except health posts. However, in line with the sustainable development goals (SDGs) and the Health Sector Transformation Plan (HSTP) 2016–2020 of Ethiopia, services for the management of NCDs are being delivered at the district, zonal and regional hospitals in Ethiopia (Ethiopian Public Health Institute, 2018).

Even though hospitals are expected to provide higher specialized care to patients with NCDs and their associated medical complications, they were not qualified. The NCD services coverage was estimated at 61% in southern nations, nationalities, and peoples region (SNNPR). Of the total 89 hospitals found in the area, only 41 and 49 provided diabetes and cardiovascular disease management, respectively (Lisa & Given, 2008).

However, despite the availability of NCD-related services in these hospitals, the mortality rate due to NCDs is high, which is 29% (Endriyas et al., 2018), and the uptake of the NCD services is low, especially in government hospitals/facilities (Shiferaw et al., 2018). Failure to optimally utilize the NCD services is associated with high morbidity and mortality. The SARA tool covers questions on essential services through service availability (whether the facility offers a variety of services for the treatment and diagnosis and readiness (whether the facilities have the items required to deliver at the visit)). However, the SARA tool is not used to fill the gaps to be filled.

Methods and materials

Study setting and population

Study area/setting

The study area was Sidama regional state which is the newly established region in Ethiopia. In the region, 708 health facilities are found viz 18 hospitals, 137 health centers, and 553 health posts.

Sidama region is located 273 km south of Addis Ababa. The region was previously a part of the Southern Nations, Nationalities, and Peoples’ Region (SNNPR) with Zonal Administration level. It has become Ethiopia’s 10th regional state in a referendum conducted in November 2019. The administrative center for Sidama is Hawassa City. According to the 2017th census conducted by the Central Statistics Agency (CSA), the region’s total population was 2,954,136 with an area of 6,538 km2. Sidama has a population density of 452/km2 with an average household size of 4.99 persons. Of the population, 5.51% are urban inhabitants, and 0.18% is pastoralists. A large area of the Sidama land produces coffee, which is the major cash crop in the region. Enset (Ensetventricosum) is the single most crucial root crop grown in the study area and is a staple food on which the bulk of the population depends heavily for survival (Central Statistical Authority (CSA), 2012).

The study was conducted from February 2021–July 2021.

Population

Source population

The source for the study is primary health care units (health centers, primary hospitals, and general hospitals) found in the region.

Inclusion and exclusion criteria

Inclusion criteria

All health centers,

Primary hospitals

General hospitals

Exclusion criteria

Health centers which were closed during the survey time

Unwilling heads and NCD focal persons of facilities were excluded from the study

Study design, study setting, and sampling technique

The study design

This study employed a facility-based cross-sectional study design. The study used the WHO Service Availability and Readiness Assessment (SARA) tool to investigate the availability of services and the primary health care unit (PHCU) readiness to manage type 2 diabetes and hypertension.

Sample size

We reviewed different articles (Lemeshow et al., 1997; Anand et al., 2018; Kachimanga et al., 2017; Basu et al., 2019) to get the input required to estimate sample size and used the following formulae (Verduin, de Gans & Dhont, 1996).

n=[[(z2∗p∗q)+ME2]/[ME2+z2∗p∗q/N]]∗d

where:

n is the sample to be calculated,

Z2 is the square of the normal deviate at the required confidence level,

ME is the margin of error,

p is the anticipated proportion of facilities with the attribute of interest,

q is 1 − p, and

d is the design effect

n = [[(1.962 * 0.47 * 0.53) + 0.152]/[0.152 + (1.962 * 0.47 * 0.53)/708]] * 2

n = [[0.97944256]/[0.0238516138]] * 2

n = [41.0639954266] * 2

n = 82

A total of 82 PHCU were included in the study.

Sampling technique

As described in Fig. 1 below, a multistage cluster sampling technique was employed. After getting the list of clusters (woredas) in the region, the study woredas were selected to draw the selected health facilities, and a simple random sample procedure was used.

Figure 1 Sample facilities recruited for SARA.

As described in Fig. 1 above, according to the Sidama regional health bureau census of health facilities, 2,020 health facilities were selected by using a simple random sampling method as follows.

Data collection tools, data collection, and data management

The WHO SARA questionnaire was used to assess the readiness of the PHCU to manage T2 DM and HPN (World Health Organization, 2013). As presented in Table 1 the tool consisted of four basic domains, basic amenities, staff training and guidelines, essential diagnostic equipment, and basic medicines.

Table 1 Domains and indicators used for assessing the readiness of primary health care units to manage diabetes and hypertension.

Domain	Indicator	Data collection and description	
Basic amenities	Sanitation facilities	Observed availability of functioning	
Communication equipment’s	Sanitation facilities	
Consultation room	Communication equipment’s	
Water source, power source, emergency transportation and computer with internet	Consultation room
Water source, power source, emergency transportation and computer with internet	
Staff training and guidelines	Guidelines for the diagnosis and management of DM and HPN	Observed availability of National Standard Treatment hard copy of Guidelines for DM and HPN.	
Staff trained in the diagnosis and management of DM and HPN	Observed availability of at least one trained staff providing	
	DM, HPN services who had completed refresher courses for the diagnosis and management of DM, HPN.	
Basic diagnostic equipment	DM: weight scale, height scale and measuring tape and glucometer.	Observed availability of functioning weight scale, height scale and glucometer with test strips and urine dipstick protein and ketone	
HPN: stethoscope and blood pressure (BP) apparatus	Observed availability of functioning digital or manual	
	BP apparatus in the facility	
Basic medicines	DM: metformin, glibenclamide and insulin injection.	Observed availability of a first-line regimen that had not expired; metformin (for obese patients), glibenclamide (for non-obese patients) and insulin injection for DM	
HPN: thiazide diuretics, ACE inhibitors, beta
blockers and furosemide	Observed availability of at least one type of thiazide diuretics as a first-line regimen for hypertension atleast one type of (ACE inhibitor+beta blocker furosemide).	

The data were collected during the working days (Monday–Friday) while the health facilities were in their usual routine activities. The data collectors with the primary investigator (PI) interviewed the person in charge of the NCD unit at the primary and general hospitals and the heads of the health centers. The study’s purpose was presented to the research participants, and they were given the opportunity to participate and explain why the data were collected. The interviewees who accepted the research aims and signed a written informed consent were interviewed. This study was approved by the Institutional Review Board of Hawassa University (IRB), Ethiopia, and number IRB 072/13.

The above-mentioned domains and indicators were drawn from the guidelines of the Ethiopian NCD clinical management manual for NCDs (Walelgne, Yadeta & Feleke, 2016).

DATA processing and analysis

The Primary Investigator processed the data by using SPSS version 24. Descriptive statistics like; frequency and percentage, inferential statistics like; the chi-square test, and a logistic regression model was used to analyze the data.

The PI provided two days of training to the selected data collectors, which focused on data collection techniques, tools, and research ethics. Supervisors with at least an MSc in Public Health are trained to supervise the data collection process and the investigator.

Result

Background characteristics of surveyed health facilities

This section presents results from the facility inventory assessed by SARA. Two main categories of results are presented. These are: (1) the descriptive part for the availability of basic amenities, guidelines, and staff trained, diagnostic materials and medicines for both DM and HPN management; (2) the association between the availability of guidelines, staff trained, diagnostic materials, and medicines with the readiness of the PHCU to manage both DM and HPN.

As can be seen in Table 2, the majority of the 82 healthcare facilities included in the survey, 63 (76.8%), were health centers, comprising 46 (56%) situated in rural areas and 17 (23%) in urban areas. More than half of the primary hospitals, eight (9.7%) were located in the woreda towns. In comparison, only five (6%) of the primary hospitals were located in a rural setting. Two of the three general public hospitals were located in rural, and only one was in urban.

Table 2 Study facilities and their location.

Facility type	Rural facilities	Urban facilities	Total	
	n	%	n	%	n	%	
Health centers	46	56	17	23	63	76.8	
Primary hospital	5	6	8	9.7	13	16	
General hospital	2	2.4	1	1.2	3	2.4	
NGO clinics	3	3.6	0	0	3	3.6	
Total	56	68.3	26	31.7	82	100	

As presented in Fig. 2 above, basic amenities, sanitation facilities, communication equipment, consultation room, improved water source, electric power source, emergency transportation (ambulance), and computer with internet access were assessed for availability and functionality. Figure 2 shows that 62 (75.6%) of the facilities had a sanitation facility, 48 (58.5%) had communication equipment, 32 (38.9%) had a consultation room, 45 (54.9%) had improved water source, 54 (65.8%) had any concept of electrical power, 74 (90.2%) had emergency transportation at the nearby woreda office, and 42 (51.2%) of the facilities had a functioning computer with internet connectivity.

Figure 2 The availability of basic amenities in the sampled health facilities.

As clearly shown in Fig. 3 above, of the total health facilities (n = 82), only 19% of the facilities had NCD guidelines. Besides, 54% of the PHCU had a trained staff for the management of HPN, whereas only 27% of them had a trained staff for the management of DM. The existence of guidelines for diabetes mellitus and HPN varied greatly depending on the kind of facility, with health centers having lower availability than hospitals. The proportion of facilities with at least one staff who had received refresher training on NCDs was low across all facilities. Furthermore, the availability of these trained staff was significantly higher in hospitals than in health centers located in the rural (P < 0.05).

Figure 3 The availability of guidelines and trained staff for DM and HPN management.

As explained in Fig. 4; regardless of the facility type, the availability of a functioning adult weight scale and height scale was consistently high across all facilities at 87.3% and 84.2%, respectively. At the same time, measuring tape was significantly lower in all types of health facilities and their location.

Figure 4 Availability of basic screening equipment for DM.

As described in Fig. 5, the availability of blood glucose tests was relatively good. The test kit was more available in primary and general hospitals with a significantly lower proportion of functioning glucometers with testing strips found in rural health centers than urban facilities.

Figure 5 The availability of basic diagnostic materials for DM management.

Essential medicines and commodities for the management of HPN and DM

We evaluated the availability of medicines and commodities to manage both HPN and DM based on SARA (Figs. 6A and 6B). Consequently, the availability of angiotensin-converting enzyme (ACE) inhibitor was relatively higher 41 (50%) than thiazide diuretic 35 (42.7%), beta-blocker 15 (18.3%) calcium channel blockers 34 (41.4%). The availability of antihypertensive medicines was consistently lower at the rural health facilities than urban. Hence, considering the above description in Fig. 3, only 29% and 28% of the PHCU were identified as ready to manage HPN and DM, respectively. The overall availability of anti-diabetic agents was, metformin tab/cap, 30 (36.5%), glibenclamide tablet 32 (39%), regular insulin 25 (30.5%) glucose 40%, 49 (59.8%) and minimal health facilities, three (3.6%) do have gliclazide or glipizide tabs. The availability of metformin, glibenclamide, and regular insulin was consistently higher in hospitals and urban health centers than in facilities located in rural.

Figure 6 (A) The availability of basic medicines for the management of HPN and (B) the availability of basic medicines for the management of DM.

Determinants of the readiness of the health facilities to manage DM and HPN cases

In this topic, we tried to see (1) the association of facility type and location, with the availability of guidelines and at least one staff trained to manage type II DM and HPN. (2) the association of basic diagnostic equipment, essential medicines, and commodities needed for the management of HPN and DM with the readiness of the PHCU.

The readiness of the PHCU was evaluated based on the criteria developed by the WHO-SARA. Therefore, their readiness was assessed based on three attributes, availability of treatment guidelines and availability of at least one staff trained in the management of DM and HPN, availability of basic diagnostic materials, and the availability of essential medicines for each disease. Therefore, we considered the availability of guidelines and at least one staff trained and the availability of functioning BP apparatus and stethoscope for HPN and blood glucose test with an unexpired test strip for type 2 DM as mandatory. Considering the availability of at least one type of medicine from each group (ACA inhibitors, thiazide groups, beta-blockers, and calcium channel blockers) is compulsory for HPN management. In contrast, the availability of metformin for non-obese and glibenclamide for obese patients, insulin, and 40% glucose is deemed obligatory for the management of DM. We considered the above-mentioned operational definition; the PHCUs were either ‘ready’ or ‘not ready’ to manage type 2 DM or HPN.

Readiness of the facilities for the management of HPN with the independent variables

The following bivariate tables will discuss the association of facility type and location with the availability of trained staff and the guidelines to manage DM and HPN Table 3.

Table 3 Association of facility type, facility location and the availability of guidelines and trained staff for the management of HPN and DM.

Variables	Categories	HPN management and treatment	P-value	DM management and treatment	P-value	
Ready	Not ready	Ready	Not ready	
Facility type	Health center	1,152	52	0.001**	12	51	0.001**	
Primary Hospital	76	6	7	6	
General Hospital	3	0	3	0	
Others	2	1	2	1	
Total	23	59	24	58	
Facilities location	Rural	47	9	0.001**	48	8	0.00**	
Urban	12	14	10	16	
Total	59	23	58	24	
Availability of guideline for NCD	Yes	1	14	0.000**	3	12	0.00**	
No	58	9	55	12	
Total	59	23	15	67	
Staff trained for NCD	Yes	21	22	0.000**	3	18	0.00**	
No	38	1	55	6	
Total	59	23	21	61	
Notes:

* P value ≤ 0.05.

** P value ≤ 0.01.

As described in table below, stethoscopes and BP apparatus availability at the facilities was relatively high compared to primary and general hospitals, 80 (94.5%) and 81 (97.5%), respectively. In comparison, we identified only 23 (28.0%) of the facilities with stethoscopes and BP apparatus ready for the screening and management of HPN. Due to the availability of BP apparatus and stethoscope in the majority of the health facilities, the test statistics were insignificant at P > 0.05 (1.000). The researcher also tried to triangulate the availability of the adult weight scale, stadiometer, measuring tape, blood glucose test, and urine dipstick protein and ketones with the readiness of the health facility to manage DM.

Only 24 (29.3%), 24 (29.3%), 17 (20.7%), 23 (28.0%), and 23 (28%) of the facilities had adult weight scale, stadiometer, measuring tape, blood glucose test, and urine dipstick protein and ketones respectively were ready to manage DM and HPN. The diagnostic equipment for DM, stadiometer, measuring tape, blood glucose test, and urine dipstick protein and ketones were significantly associated with the readiness of the PHCU with a P < 0.05. Because of vast availability, the adult weight scale was not associated with the readiness of the PHCU at a P > 0.348.

The following table (Table 4) describes the availability of diagnostic equipment to manage HPN and DM.

Table 4 Availability of diagnostic equipment for HPN and DM management.

Diagnostic equipment’s for HPN management and treatment	
Equipment	Category	HPN management and treatment	P-value	
Ready	Not ready	
Stethoscope	Yes	23	57	1.000	
No	0	2		
Total	23	59		
Blood pressure apparatus	Yes	23	58	1.000	
No	0	1		
Total	23	59		
Diagnostic equipment’s for DM management and treatment	
Measuring tape	Yes	17	7		
No	7	51	0.000**	
Total	24	58		
Adult weight scale	Yes	24	55	0.348	
No	0	3		
Total	24	58		
Stadio-metere	Yes	24	45	0.008*	
No	0	13		
Total	24	58		
Blood glucose test	Yes	23	18		
No	1	40	0.000**	
Total	24	58		
Urine dipstick-protein and ketone	Yes	23	20		
No	138	38	0.000**	
Total	24	24		
Notes:

* P value ≤ 0.05.

** P value ≤ 0.01.

Readiness of the facilities to manage HPN

The following tables describe the basic medicines and commodities used to determine the readiness of the PHCU to manage HPN and DM, respectively.

These were medicines and commodities to manage HPN; at least one type of ACA inhibitor, thiazide diuretics, beta-blockers, and calcium channel blockers and metformine, glibenclamide, insulin, glucose 40%, and gliczaide for the management of DM (Table 5).

Table 5 The bivariate analysis of basic medicines for the management of HPN with readiness of the PHCU.

Variables	Categories	HPN management and treatment	P-value	
Ready	Not ready	
At least one type of ACE inhibitor	Yes	17	13	0.013*	
No	6	46		
Total	23	59		
At least one type of thiazide diuretic	Yes	22	13	0.000**	
No	1	146		
Total	23	59		
At least one type of Beta blocker	Yes	12	3	0.000**	
No	11	56		
Total	23	59		
At least one type of calcium channel blockers (e.g., amlodipine)	Yes	22	12	0.000**	
No	1	48		
Total	1	59		
Notes:

* P value ≤ 0.05.

** P value ≤ 0.01.

Of the total health facilities, only 17 (20.7%), 22 (26.8%), 12 (14.6%), and 22 (26.8%) had at least one type of ACE inhibitor, a thiazide diuretic, beta-blockers, and calcium channel blockers, respectively were ready to manage HPN. The association between the availability of antihypertensive drugs and the readiness of the health facility to manage HPN was strongly significant at P < 0.05.

The researchers assessed the availability of basic DM medicines with the readiness of health facilities to manage DM (Table 6). Therefore, facilities that had 23 (28.0%) metformin, 23 (28.0%) glibenclamide 20 (24.4%), insulin regular injectable 40% 24 (29.3%), glucose, and gliclazide or glipizide tablet 2 (2.4%), was found ready to manage DM. Moreover, the availability of medicine and the facility’s readiness was strongly associated with the first four medicines at a P-value = 0.00 (<0.05). While the availability of gliclazide tablets was almost nil at all health facilities except general hospitals. Thus the readiness of the PHCU to manage DM was not significantly associated at P > 0.05.

Table 6 The bivariate analysis of basic medicines for the management of DM with readiness of the PHCU.

Variables	DM management and treatment	P-value	
	Ready	Not ready		
Metformin cap/tab	23	7	0.000	
1	51		
24	58		
Glibenclamide cap/tab	23	6	0.000	
1	49		
24	58		
Insulin regular injectable	20	5	0.000	
4	3		
24	58		
Glucose 50% injectable	24	25	0.000	
0	33		
24	58		
Gliclazide tablet or Glipizide tablet	2	1	0.0204	
22	57		
24	24		

Predictors of health facilities’ readiness for HPN management

Based on Table 7 below, the variables facility type, facility location, availability of guidelines, trained staff, ACE inhibitor thiazide groups, and beta-blockers had a significant impact (P < 0.05) on the readiness of the PHCU to manage HPN at a 0.05 level of significance.

Table 7 Factors associated with the readiness of the facilities to provide HPN management at the PHCU adjusted for confounding variables.

Variables	COR [95% CI]	AOR [95% CI]	
Facility type	
Health centre	1.0	1.0	
Primary Hospital	5.5 [1.5–19.6]*	3.6 [0.9–14.9]	
General Hospital	7,636,790,166	-	
Others	9.5 [0.8–0.113]	22.2 [1.7–297.9]*	
Location of the Facilities	
Rural	1.0	1.0	
Urban	6.0 [2.1–17.4]**	8.2 [2.4–28.5]*	
Guideline for CVD	
Not available	1.0	1.0	
Available	0.011 [0.001–0.095]**	0.005 [0.000–0.091]**	
Staff trained for HPN management	
Not trained	1.0	1.0	
Trained	0.025 [0.003–0.20]**	0.066 [0.008–0.564]*	
Stethoscope	
Available	1.0	1.0	
Not available	0.000	-	
BP apparatus	
Available	1.0	1.0	
Not available	0.000	-	
Groups of medicine	
ACE inhibitor	1.0	1.0	
Thiazide groups	0.242 [0.083–0.703]*	0.095 [0.016–0.570]	
Beta blockers	0.013 [0.002–0.105]**	0.012 [0.001–0.135]**	
Calcium channel blocker	0.049 [0.012–0.203]**	0.011 [0.001–0.149]**	
Notes:

* P value ≤ 0.05.

** P value ≤ 0.01.

Facilities located in urban were 8.2 times more likely to be ready to provide HPN management than facilities located in rural (AOR = 8.2, 95% CI [2.4–28.5]). PHCU, which had no guidelines, trained staff, antihypertensive medicines such as ACE inhibitors, thiazide groups, and beta-blockers, were 0.005, 0.066, 0.095, 0.012, and 0.011 times less likely to get ready in the management of manage HPN than those who had guidelines, trained staff, and at least one type of medicines from the group of medicines respectively. (AOR = 0.005, 95% CI [0.000–0.09], AOR = 0.066, 95% CI [0.008–0.564], AOR = 0.095, 95% CI [0.016–0.57], AOR = 0.012, 95% CI [0.001–0.135], AOR = 0.011, 95% CI [0.001–0.149]) at a P < 0.01.

Predictors of health facilities’ readiness for DM management

As displayed in Table 8 below, most of the binary logistic regression results showed a clear link between the availability of the trained staff and guidelines on NCD with the dependent variable, ‘readiness to screen and manage DM’ at P < 0.05. In the meantime, the availability of some diagnostic equipment like measuring tape, blood glucose test, urine dipstick-protein, and ketones is strongly associated with the readiness of the PHCU. Moreover, the availability of antidiabetic medications like Metformin cap/tab, Glibenclamide, cap/tab, Insulin regular injectable were strongly associated with the readiness of the health facility too. In comparison, the type of the facility, availability of adult weight scale, BP apparatus, stadiometer, glucose 50% injectable, gliclazide tablet, or glipizide tablet was not associated with the readiness of the PHCU at P > 0.05 and CI of 95%.

Table 8 Factors associated with the readiness of the facilities to provide DM management at the PHCU adjusted for confounding variables.

Variables	COR [95% CI]	AOR [95% CI]	
Facility type	
Health centre	1.0	1.0	
Primary Hospital	4.958 [1.4–17.5]*	2.2 [0.5–9.5]	
General Hospital	–	–	
Others	8.5 [0.7–101.6]	11.9 [0.8–0.186.0]	
Facilities location	
Rural	1.0	1.0	
Urban	9.6 [3.2–28.5]**	11.1 [3.13–39.36]**	
Guideline for NCD	
Not available	1.0	1.0	
Available	0.055 [0.013–0.22]**	0.31 [0.004–0.231]**	
Staff trained for DM management	
Trained	1.0	1.0	
Not trained	0.18 [0.004–0.08]**	0.055 [0.009–0.336]*	
Adult weight scale	
Not available	1	1	
Available	0.000	-	
Measuring tape (ref.)	
Not available	1.0	1.0	
Available	0.057 [0.017–0.18]**	0.055 [0.009–0.336]**	
Stadio-meter	
Available	1	1	
Not available	-	-	
Blood glucose test	
Not available	1	1	
Available	0.04 [0.002–0.156]**	0.044 [0.005–0.339]**	
Urine protein and ketone	
Not available	1	1	
Available	0.023 [0.003–0.182]**	0.054 [0.006–0.472]**	
Metformin	
Available	1	1	
Not available	0.006 [0.001–0.051]**	0.003 [0.000–0.066]**	
Glibenclamid	
Not available	1	1	
Available	0.008 [0.001–0.67]	0.008 [0.001–0.091]	
Insulin	
Not available	1	1	
Available	0.19 [0.005–0.77]	0.018 [0.003–0.121]**	
Glucose 40%	
Not available	1	1	
Available	-	-	
Gliclazide/Glipizide	
Not available	1	1	
Available	0.193 [0.017–2.237]	0.459 [0.011–19.545]	
Notes:

* P value ≤ 0.05.

** P value ≤ 0.01.

Discussion

This study assessed the readiness of PHCUs found in Sidama regional states to manage DM and HPN. The study identified significant gaps in the availability of basic amenities, staff training and guidelines, diagnostic equipment and materials, medicines, and commodities. Although, there is variability among the different types of health facilities and where the facilities are located to manage DM and HPN.

Overall, the study found disparities in the readiness between the type of health facilities and their service. Hospitals provide a higher level of service for their communities than health centers.

When looking at individual categories of the WHO-SARA (availability of trained personnel and guidelines, basic diagnostic materials, and basic medicines and commodities) according to the type of the facility, health centers had lower availability of guidelines and trained staff to manage DM HPN. This finding is consistent with research undertaken in Uganda and Tanzania (Paromita et al., 2021; Rogers et al., 2018; Katende et al., 2015). The unavailability of guidelines in the health centers and low availability in the primary and general hospitals is evidence of the problems related to early detection of DM and HPN (Walelgne, Yadeta & Feleke, 2016; Paromita et al., 2021).

The study also revealed that the lower proportion of health care providers trained for DM and HPN in all types of health facilities, and the problem is serious at the rural health facility level. This finding is consistent with previous research conducted in Uganda, Nigeria, sub-Saharan countries and Bangladesh (Paromita et al., 2021; Rogers et al., 2018; Rawal et al., 2019; Cárdenas et al., 2021). The similarity could be the financial problem that the countries are facing, the attention given to NCDs, or the country’s policy toward NCD (Juma et al., 2016; Wang et al., 2016).

The availability of basic diagnostic materials for all types of DM and HPN significantly impacts the early identification and management of the problems (World Health Organization, 2015). This study revealed a significant difference in the availability of diagnostic materials between health facilities. Concerning the availability of essential risk factor screening tools for DM, the availability of weight scale and height scale among the health facilities were high. Our finding is consistent with research conducted in Tanzania and Cambodia (Bintabara & Ngajilo, 2020; Jacobs et al., 2016). The reason for the availability of the material in the facilities may be elucidated because the weight and height scales are attached and used for other purposes like maternal and child care (MCH). Besides, facility type had an excellent contribution to the availability of glucometers with their testing strips. We found that similar studies conducted in Tanzania and Uganda have identified a higher proportion of glucometer with its strip found in hospitals than in health centers and dispensaries (Paromita et al., 2021; Bintabara & Ngajilo, 2020). The present study also revealed that a high proportion of blood pressure measuring apparatus and stethoscopes as essential diagnostic equipment for HPN screening and management was available in the majority of the health facility. The study also confirmed no significant difference in the availability of the machine and stethoscope in facility type. Studies conducted in Ethiopia, Nigeria, and Uganda confirmed similarities between the results (Bintabara & Ngajilo, 2020; Vladislav et al., 2019; Bollyky, 2013). This finding may be because BP machines and stethoscopes are considered basic materials to take the patient’s vital signs at the triage unit. It can use for antenatal, laboring, and postnatal periods to screen pre-eclampsia, eclampsia, and postnatal complications.

Access to effective treatment for NCDs is a vital step in reducing the risk and burden of the disease (Bollyky, 2013). However, services remained lower according to the type of facilities in many LMIC (Bintabara & Mpondo, 2018; Legese & Tadiwos, 2020). Overall, the availability of metformin, glibenclamide, and injectable insulin reported in this study was consistently higher in hospitals than in health centers. Similar investigations have been carried out in various parts of the world. This study was in line with studies conducted by different researchers (Bintabara & Ngajilo, 2020; Vladislav et al., 2019; Faatoese et al., 2011). In addition, relatively half of the health facilities have ACE inhibitors. However, a lower proportion of facilities have at least one type of Hydrochlorothiazide, beta-blockers, and calcium channel blockers at the lower levels of the health facilities (health centers). This finding agrees with other African countries’ reports like Cameroon, Tanzania, Zambia, Uganda, and Nigeria (Ademe, Tebeje & Molla, 2016; Adinan et al., 2019; Shiroya et al., 2021; Paromita et al., 2021).

The disparities in the availability of basic screening/diagnostic materials and treatment medicines and commodities among facility types in the current study may be due to several problems like lack of trained health care providers, especially at the lower levels of health facilities (health centers), the unavailability of medicines and commodities at the official distributor, a complex procurement process arranged by the government (Ademe, Tebeje & Molla, 2016). Inadequate budget allocation and insufficient ordering of the NCD medicines by the health care facility, and lack of appropriate knowledge of the healthcare providers found in the rural health centers worsened the problem (Chang et al., 2019). Health centers are expected to provide NCD services for their community members. In contrast, medicines and medical commodities essential for NCD screening and treatment were not available at the health center level.

Therefore, we recommend that essential NCD screening and management medicines and commodities be available at the lower and rural health centers to consider the rural communities.

The differences in the availability of diagnostic and treatment commodities have a significant impact on the readiness of the health facility to provide early detection of DM and HPN. The variations in availability of services based on; the availability of trained staff and guidelines, the availability of basic diagnostic materials for both DM and HPN, and the availability of essential medicines and commodities have a significant impact on the readiness of facilities to provide early detection and management of NCD services. At the same time, Ethiopian health policies and guidelines necessitate all facilities, including health centers, to screen for major NCDs (Agunga, 2018). Whereas this study identified that facilities located in rural are less ready than those found in urban. Lesser readiness is identified at health centers than in hospitals (primary and general). This may be because of the lower requesting ability of the health care providers at the health centers than hospitals and the lower level of the academic background of the health care providers at the health center level than hospitals.

According to our study, facilities located in urban were 8.2 times more likely to be ready to provide HPN management than facilities located in rural. This outcome is consistent with findings from earlier Tanzanian (Adinan et al., 2019), Zambian, and Malawian investigations (Shiroya et al., 2021). The good reason for not providing DM and HPN at the rural health facilities could be improper supportive supervision from the nearby health offices, a little concern for NCDs, and a lack of political commitment (Tesema et al., 2021; Bausch et al., 2021; Agunga, 2018).

The study also found that primary hospitals and NGO clinics were 3.6 and 22.3 times more likely to handle HPN and DM than health centers. This finding is consistent with recent studies conducted in Tanzania (Adinan et al., 2019) and Ethiopia (Tesema et al., 2021). The projected reason for this could be lack of trained and skilled health care providers, lack of diagnostic equipment and materials at the health centers, lack of political and leadership commitment (Tesema et al., 2021; Agunga, 2018; Isadru, Nanyonga & Alege, 2021).

PHCUs that do not have guidelines, trained personnel, and antihypertensive and anti-diabetic drugs were 0.005, 0.066, 0.095, 0.012, and 0.011 times less likely to manage HPN than those with guidelines, trained staff, and at least one type of medicine from the category of medicines. Our result is also in-line with studies conducted in other LMIC and sub-Saharan African countries (Adinan et al., 2019; Isadru, Nanyonga & Alege, 2021; Ghimire et al., 2020). The predicted cause for this could be a lack of political will and leadership, loose policy implementation on NCD prevention, and a lack of knowledge on the severity of the disease (Juma et al., 2016; Tesema et al., 2021; Agunga, 2018).

This study offers valuable policy inferences to progress NCD services at all facilities regardless of their location. The recommendations include an improved country-wide obligation to offer NCD services in all facilities as stipulated in the existing policies and guidelines. In addition, tactics must be developed to facilitate better policy results incorporating local context factors such as political will, allocating resources, and so forth, and assignment of qualified health care providers (at least a health officer or BSc nurse) at the rural health centers. MoH should revise policies for procurement of medicines and medical supplies to strengthen lower-level facilities, particularly health centers located in the rural areas serving the majority of the Ethiopian population. Also, efforts should be made to meet the voluntary target set by the Global Action Plan on NCDs of an 80% availability of affordable basic technologies and medicines required to treat major NCDs in both public and private facilities (Menezes, Lopes & de Nogueira, 2016). Through this, we can reduce premature disabilities and death in the country.

The main strength of this study is the use of primary data directly from the study setting and investigating the current availability and functionality of the screening equipment, medicines, and commodities. The researchers created the outcome variables based on indicators identified from the WHO-SARA manual, which desired to be available in the national guidelines of Ethiopia for the prevention and treatment of NCDs.

Limitation of the study

The drawback of this study was its cross-sectional nature. Service availability and readiness measures may not detect the quality and experience of service delivery, which has usually become a higher contributor to health loss in LMICs related to service availability alone.

Implications of the result

The outcome of this study has broad implications for the NCD response in Ethiopia. First, this study adds to the body of knowledge relating to primary health care units’ readiness to manage diabetes and hypertension in particular and NCDs in general. This study adds depth to the understanding of the policymakers how to address communities who reside in the rural setting. Also, it is beneficial to mitigate the risk of undiagnosed NCD to ensure readiness for NCD services. Facilities in rural settings lack trained staff, guidelines, screening and diagnostic materials, and medicines and medical commodities to manage DM and HPN. As a result, the probability of early detection will deteriorate so that the possibility of getting severe complications such as macro and microvascular damage will be higher. The health costs will put the patient into an unbearable economic burden. Therefore, making PHCU ready to manage DM and HPN should be the primary focus of the government of Ethiopia.

Conclusion

In summary, this study provided tangible evidence of the disparities in readiness of the PHCU in the early detection and management of DM and HPN. The current results identified a gap in the availability of trained staff members and treatment guidelines, essential diagnostic equipment, and medicines; the researchers observed comparatively poor and deprived readiness to offer HPN and DM services at the lower level and rural health facilities (health centers).

Recommendations

Equipping the lower-level health facilities with all the required will improve the quality of service provided for NCD patients. The higher levels of health care are overloaded with the management of routine cases. There should be improvements to be geared towards six main areas: (i) provision of refresher training for the health care providers at the rural health centers, (ii) provision of standard guidelines, (iii) procurement and re-filling of essential diagnostic equipment and medicines; (iv) critical follow-up schedule should be arranged for NCDs by the woreda health offices; (v) professionals with at least first degree in public health or nursing should be assigned at the rural health facilities; (vi) nearby primary hospitals should schedule an outreach program, and they should provide on-site training for the health care providers of the rural health centers.

Supplemental Information

Supplemental Information 1 SPSS dataset of facilities.

Click here for additional data file.

Supplemental Information 2 Consent form.

Click here for additional data file.

Additional Information and Declarations

Competing Interests

Author Contributions

Ethics

Data Availability

The authors declare that they have no competing interests.

Tigist Kebede Mulugeta conceived and designed the experiments, performed the experiments, analyzed the data, prepared figures and/or tables, authored or reviewed drafts of the article, and approved the final draft.

Dejene Hailu Kassa performed the experiments, prepared figures and/or tables, authored or reviewed drafts of the article, and approved the final draft.

The following information was supplied relating to ethical approvals (i.e., approving body and any reference numbers):

The Hawassa University College of Medcine and Health Sciences Institutional Review Board approved the study (IRB/072/13).

The following information was supplied regarding data availability:

The raw data is available in the Supplemental File.

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
