# Peer review of "Readiness of the primary health care units and associated factors for the management of hypertension and type II diabetes mellitus in Sidama, Ethiopia"

_PeerJ, doi:10.7717/peerj.13797_

## Round 0.1 · original submission · Minor Revisions

Please, look rigorously at the indication of corrections done by reviewers. Regards.

·

Basic reporting

NO COMMENTS

Experimental design

NO COMMENTS

Validity of the findings

NO COMMENTS

Additional comments

GOOD TEXT, WITH RELEVANT RESULTS AND SATISFACTORY DATA

·

Basic reporting

The article is written in English, however it needs improvement to ensure that an international audience can clearly understand its text. Some examples where language could be improved include lines 18, 24, 25, 40, 41, 78, 79, 113, 114, 116 (correct and observe the rule for writing the scientific name), 171 (identify what is PI), 268-271, 284-286, 292-293, 341-344 (double text), 368-369, 409-411 – current phrasing makes comprehension difficult.

The acronyms DM, HPN and DALYs (lines 20 and 21, 57) appear in the text without prior identification. Although in a brief reading it is possible to identify their meanings, I think it is important to carry out the prior identification of the acronyms used.

The text includes sufficient introduction and background to demonstrate how the work fits into the broader field of knowledge.

The literature is referenced and relevant, however there are problems to be corrected. As the journal was corrected, the following stand out: references and references do not follow the pattern suggested by the journal o for reference is linked between lines 5 5 no; as references number 4, 7, 17, 30 and 36 are not correct.

The structure of the text presented is in accordance with the standards of the journal, except for citations and references, as mentioned above.

The figures and tables are relevant, however they need a better description, as for example, in figure 2, where in a brief observation it is not clear the meaning of the orange bar.

Experimental design

The submitted work is within the scope of the journal.

The question the research aims to answer is well defined, relevant and meaningful.

The submitted article presents research with satisfactory technical and ethical standards.

The methods are described with details and information, allowing their replication by other researchers.

Validity of the findings

The article brings relevant contributions to the community where the study was carried out. Similar works can be carried out in other locations (cities, states, countries) using the same modalities of diseases or other diseases of importance in public health.

The data on which the conclusions are based have been provided, where these data can be considered as robust, statistically sound and controlled.

Based on the reading of the article, it can be said that the conclusions were formulated and linked to the original research question and limited to the results obtained with the research.

·

Basic reporting

The work has a good scientific and academic. The work will be able to support the medical community and also towards changes in units in the practical field for medical students and the health area in general.

But we need to make some observations in the study and in its elaboration of result and discussion.

1: Both in the introduction and in the discussion, as many as can cite and compare their results and their hypotheses with articles that compare their collections in other continents and that have similar objectives.

Williams JS, Walker RJ, Smalls BL, Hill R, Egede LE. Patient-Centered Care, Glycemic Control, Diabetes Self-Care, and Quality of Life in Adults with Type 2 Diabetes. Diabetes Technol Ther. 2016 Oct;18(10):644-649. doi: 10.1089/dia.2016.0079. Epub 2016 Aug 19. PMID: 27541872; PMCID: PMC5069713.

de Oliveira KC, Zanetti ML. Conhecimento e atitude de usuários com diabetes mellitus em um serviço de atenção básica à saúde [Knowledge and attitudes of patients with diabetes mellitus in a primary health care system]. Rev Esc Enferm USP. 2011 Aug;45(4):862-8. Portuguese. doi: 10.1590/s0080-62342011000400010. PMID: 21876885.

Hersson-Edery F, Reoch J, Gagnon J. The Quebec Diabetes Empowerment Group Program: Program Description and Considerations Regarding Feasibility and Acceptability of Implementation in Primary Health Care Settings. Front Nutr. 2021 Mar 12;8:621238. doi: 10.3389/fnut.2021.621238. PMID: 33816538; PMCID: PMC8014038.

2: In the introduction, I could cite works that show the reality of Diabetes and Hypertension not only worldwide, but also in different continents, making a connection with the care units in these locations.
3: In the results
In the results, it would be important to review the formatting of tables and graphs (figures). In the alignment as in table 3 (P-Valeu) and (Not ready, not ready).
4: Discussion
The discussion is appropriate for the purpose of the study, but it remains to show and compare the results of the work with other research carried out on other continents and in other regions with similar realities and also with different realities of health units in central and petrified (rural) regions. . This would make the work more important showing changes that can be carried out by public policies and for the academic environment for practical health courses in their field.

Experimental design

The experimental design is clear and well defined and is of great importance to the scientific community, academia and the general population. The research shows results from a region that had not yet been demonstrated and that can bring changes to that region and serve for new research in the area.

Validity of the findings

The work and its results have a relevant impact showing the situation of primary health care units and associated factors for the management of hypertension and type II diabetes mellitus in a region that has not been described before and that can be replicated by other researchers in other regions. region and that also shows a reality that can be compared with studies carried out in other continents with similar social reality and also different with different economic situation.

Additional comments

I have no additional comments to those made during the description of the manuscript evaluation.

---

## Round 0.2 · accepted · Accept

Dear authors, congratulations.